# The Clinical Significance of Anaerobic Coverage in the Antibiotic Treatment of Aspiration Pneumonia: A Systematic Review and Meta-Analysis

**DOI:** 10.3390/jcm12051992

**Published:** 2023-03-02

**Authors:** Yuki Yoshimatsu, Masaharu Aga, Kosaku Komiya, Shusaku Haranaga, Yuka Numata, Makoto Miki, Futoshi Higa, Kazuyoshi Senda, Shinji Teramoto

**Affiliations:** 1Elderly Care, Queen Elizabeth Hospital, Lewisham and Greenwich NHS Trust, London SE18 4QH, UK; 2Centre for Exercise Activity and Rehabilitation, School of Human Sciences, University of Greenwich, London SE9 2HB, UK; 3Department of Respiratory Medicine, Yokohama Municipal Citizen’s Hospital, Yokohama 221-0855, Japan; 4Respiratory Medicine and Infectious Diseases, Oita University Faculty of Medicine, Yufu 879-5593, Japan; 5Comprehensive Health Professions Education Center, University Hospital, University of the Ryukyus, Okinawa 903-0125, Japan; 6Department of Respiratory Medicine, Nagaoka Red Cross Hospital, Nagaoka 940-2085, Japan; 7Department of Respiratory Medicine, Japanese Red Cross Sendai Hospital, Sendai 982-0801, Japan; 8Department of Respiratory Medicine, National Hospital Organization Okinawa National Hospital, Okinawa 901-2214, Japan; 9Department Pharmacy, Kinjo Gakuin University, Nagoya 463-8521, Japan; 10Department of Respiratory Medicine, Tokyo Medical University Hachioji Medical Center, Tokyo 160-0023, Japan

**Keywords:** dysphagia, swallowing impairment, pneumonia, anaerobe, anaerobic coverage, antibiotic, treatment

## Abstract

Introduction: Aspiration pneumonia is increasingly recognised as a common condition. While antibiotics covering anaerobes are thought to be necessary based on old studies reporting anaerobes as causative organisms, recent studies suggest that it may not necessarily benefit prognosis, or even be harmful. Clinical practice should be based on current data reflecting the shift in causative bacteria. The aim of this review was to investigate whether anaerobic coverage is recommended in the treatment of aspiration pneumonia. Methods: A systematic review and meta-analysis of studies comparing antibiotics with and without anaerobic coverage in the treatment of aspiration pneumonia was performed. The main outcome studied was mortality. Additional outcomes were resolution of pneumonia, development of resistant bacteria, length of stay, recurrence, and adverse effects. The Preferred Reporting Items for Systematic reviews and Meta-Analyses (PRISMA) guidelines were followed. Results: From an initial 2523 publications, one randomised control trial and two observational studies were selected. The studies did not show a clear benefit of anaerobic coverage. Upon meta-analysis, there was no benefit of anaerobic coverage in improving mortality (Odds ratio 1.23, 95% CI 0.67–2.25). Studies reporting resolution of pneumonia, length of hospital stay, recurrence of pneumonia, and adverse effects showed no benefit of anaerobic coverage. The development of resistant bacteria was not discussed in these studies. Conclusion: In the current review, there are insufficient data to assess the necessity of anaerobic coverage in the antibiotic treatment of aspiration pneumonia. Further studies are needed to determine which cases require anaerobic coverage, if any.

## 1. Introduction

Aspiration pneumonia has become a leading cause of hospitalisation and death in adults. It represents a major socioeconomic burden worldwide, accounting for up to 90% of pneumonia in the older population [1]. Within community-acquired and hospital-acquired pneumonia, aspiration pneumonia is a subtype known to have a poor prognosis [2]. Therefore, it is crucial to investigate the current optimal management of aspiration pneumonia.

Anaerobic bacteria have been thought to play a major role in the pathogenesis of aspiration pneumonia. This was particularly true in the 1970s [3,4,5,6,7], when several reports identified anaerobes as the causative organisms, and new antibiotics were developed to treat them. As a result of these findings, it became common practice to consider routine anaerobic coverage in patients suspected of having aspiration pneumonia [8].

However, recent studies suggest that anaerobic coverage may not necessarily improve clinical outcomes. A shift in the bacteria commonly associated with community-acquired pneumonia (CAP) and hospital-acquired pneumonia (HAP) has been reported, with fewer anaerobes identified [3,8,9]. Recent guidelines have taken these findings into account and do not recommend the routine coverage of anaerobic pathogens in the treatment of aspiration pneumonia [10,11].

As a result of these changes, it cannot be assumed that the optimal routine antibiotic treatment for aspiration pneumonia is to cover anaerobes. There is evidence that the routine usage of anaerobic coverage may not only be non-beneficial, but also potentially harmful [12,13]. The unnecessary use of broad-spectrum antibiotics must be avoided in view of future resistance, adverse effects and healthcare costs.

There have been review articles on aspiration pneumonia, providing overviews on their pathology and management [9,14,15]. These reviews have all commented on the shift in the role of anaerobes in aspiration pneumonia over the years, and questioned the routine usage of antibiotics that cover anaerobic organisms. However, to our knowledge, no formal systematic review has been published comparing clinical outcomes with or without anaerobic coverage in the treatment of aspiration pneumonia. Clinical practice and guideline updates should reflect the most recent evidence available. Therefore, we performed a systematic review of the literature to answer the question: “Is anaerobic coverage recommended in the treatment of aspiration pneumonia?”.

## 2. Materials and Methods

A systematic review and meta-analysis of the scientific literature on the clinical significance of antibiotics with anaerobic coverage compared to antibiotics without anaerobic coverage in the treatment of aspiration pneumonia was performed. The Preferred Reporting Items for Systematic reviews and Meta-Analyses (PRISMA) guidelines were followed [16]. The protocol was registered to Prospero before initiation of the study (registration number: CRD42022358664) and can be found at the following URL: https://www.crd.york.ac.uk/prospero/display_record.php?RecordID=358664, accessed on 15 September 2022.

Patients were adults aged 18 years or older with a diagnosis of aspiration pneumonia, necrotising pneumonia or lung abscess. We added necrotising pneumonia and lung abscess so we do not exclude any potentially relevant studies, as aspiration pneumonia is still a variable term. The intervention was antimicrobial treatment covering anaerobic organisms. The control was antimicrobial treatment without coverage of anaerobic organisms. The main outcome studied was mortality, and other outcomes consisted of resolution of pneumonia, development of resistant bacteria, length of hospital stay, recurrence of pneumonia, and adverse effects. The types of studies included were primary studies published in a peer-reviewed journal. Studies from any setting and any year were included. All non-English literature, unpublished material, study protocols, conference abstracts, and book chapters were excluded to maintain the scientific quality of the review. Reviews were also excluded as they are not primary studies.

The databases searched were PubMed (https://pubmed.ncbi.nlm.nih.gov/, accessed on 15 September 2022) and Cochrane Library (https://www.cochranelibrary.com/, accessed on 15 September 2022). The search strategy was developed in PubMed and then subsequently translated for the Cochrane Library. Full strategies are provided in the Section A.1 and Section A.2. We searched for ‘aspiration pneumonia’ and ‘treatment’ using both controlled vocabulary, such as MeSH terms, and natural language terms for their synonyms. The search strategy was developed with broad terms to ensure all relevant articles would be detected in the database search. We excluded guidelines, meta-analyses, reviews, and case reports. The search was conducted on 7 September 2022. Duplicates were removed before screening using Rayyan duplicate identification strategies.

Identified studies were independently reviewed by two of the authors (Y.Y. and M.A.), and decisions were recorded using Rayyan. Disagreements were resolved by discussion and, where necessary, by review by two other authors (S.H. and Y.N.).

Inclusion criteria were original papers comparing antibacterial treatment with and without anaerobic coverage in adults (aged 18 years and older) diagnosed with aspiration pneumonia, necrotising pneumonia, or lung abscess. Exclusion criteria were reviews, case reports, editorials, conference papers, children, animals, in vitro studies, prophylactic antibiotics, and non-systemic routes of administration. Reviews were excluded from the study, but their references were searched for relevant studies. Manual searches of the reference lists of relevant guidelines [10,11,17], included studies, and other relevant publications [9,18,19] were also performed.

A data extraction form was designed to extract study characteristics and outcomes. Two reviewers (Y.Y. and M.A.) independently extracted data from eligible publications independently. The extracted data were compared, and any discrepancies were resolved by discussion between them and two other reviewers (S.H. and Y.N.). No automated tools were used. Data (odds ratio) on the primary outcome (mortality) and secondary outcomes (clinical cure rate, development of resistant bacteria, length of hospital stay, recurrence of pneumonia, and rate of adverse effects) were extracted. We also extracted information on the characteristics of the eligible studies and outcomes as follows: author, year, source of publication, sample size, sample/participant characteristics. If necessary, the authors of the publications were contacted.

Meta-analysis was performed using ReviewManager (Revman) (London, UK) for outcomes for which two or more studies provided data. For other outcomes for which only one study provided data, extracted data are presented and summarised descriptively.

The risk of bias of the observational studies [20,21] was assessed using the Newcastle Ottawa Scale (NOS) [22]. The NOS was also used to assess the randomised control trial (RCT) [23] for outcomes reported in two or more studies (mortality and clinical cure rate), to ensure consistency within outcomes. The Cohort Studies version of the scale was chosen to assess studies for subject selection, cohort comparability, and outcomes. The Cochrane Risk of Bias tool (RoB 2) was used for outcomes where only RCTs were included [24]. Two reviewers (Y.Y. and S.A.) independently assessed the risk of bias for each study, and any discrepancies were resolved by discussion.

## 3. Results

A total of 2728 studies were identified through database and manual searches. After removing 205 duplicates, 2523 reports were screened on their titles and abstracts, of which 2519 were excluded (Figure 1). The reasons for exclusion at the screening stage were: wrong population (*n* = 1437), wrong publication type (*n* = 407), wrong intervention (*n* = 176), background article (*n* = 168), wrong study design (*n* = 147), wrong language (*n* = 144), and wrong outcome (*n* = 40). Of the four studies that underwent full-text review, one was excluded due to incorrect study design [25]. Finally, three papers were included in the final analysis [20,21,23]. The study selection process is shown in Figure 1, according to the PRISMA methodology [16].

Of the three included studies, one was an RCT [23], and two were prospective observational studies [20,21]. A total of 941 subjects were included. All studies were conducted in Japan. All included studies had a mean/median age over 77 years. Their diagnoses were pneumonia with aspiration-related risk factors [21,23], or aspiration pneumonia within the NHCAP group B [20]. There was no mention of necrotising pneumonia or lung abscesses in the three studies. The characteristics of the studies are shown in Table 1. The propensity score-matched data by Hasegawa, et al. [21] were further analysed with multiple imputation by employed chain equations, and the data were not presented as integers. Therefore, raw data were used for meta-analysis to match the data in the two other studies.

Results for mortality and clinical cure rates are shown in Table 2 and Figure 2. The primary outcome and mortality were reported in all 3 studies; Oi et al. [23] reported 30-day mortality, whereas Hasegawa et al. [21] and Marumo et al. [20] reported in-hospital mortality. Overall, mortality was low and there was no significant mortality benefit in the anaerobic coverage group compared with the control group. Mortality was 9.9% (56/567) in the anaerobic coverage group, and 8.0% (30/374) in the control group (odds ratio (OR) 1.24, 95% confidence interval (CI) 0.70, 2.18).

The clinical cure rate from two studies [20,23] showed no significant benefit of anaerobic coverage; the results were 79.6% (133/167) for the anaerobic coverage group, and 78.1% (107/137) for the control group, using the intention to treat analysis (OR 1.31, 95% CI 0.74, 2.33).

Length of hospital stay was reported in only one study [20]; 22.3 ± 7.3 days in the anaerobic coverage group, and 20.5 ± 8.1 days in the control group, with no significant difference (*p* = 0.654). Hasegawa et al. [21] reported the ’28-day hospital-free days’ as a substitute for length of stay, which was significantly shorter in the anaerobic coverage group than in the control group (11 vs. 9 days; *p* = 0.005).

The rate of pneumonia recurrence was reported in one RCT [23], and it was 5.8% (5/86) in the anaerobic coverage group, and 2.0% (2/101) in the control group, using the intention to treat analysis (OR 1.97, 95% CI 0.46, 8.48).

The rate of adverse effects was also reported in one RCT [23] only, in which the rate was 22.0% (18/82) in the anaerobic coverage group, and 25.5% (24/94) in the control group, using the validated per-protocol analysis (OR 0.82, 95% CI 0.41, 1.65). No serious antibiotic-related events were reported.

The rate of development of resistant bacteria was not reported in any of the three studies.

The risk of bias assessment using the NOS is shown in Table 3. All studies were rated low in the representativeness of the expressed cohort, as they had variable definitions of aspiration pneumonia and were limited to certain severity groups. Otherwise, they were generally graded well for most of the criteria for subject selection, cohort comparability, and outcome. For the outcomes for which only one RCT was included (recurrence rate and adverse effects), the risk of bias was assessed as ‘low risk’ using the RoB 2. However, as all studies were conducted in the acute setting in Japan, with mostly inpatients, this raises a concern regarding external validity. Therefore, it can be concluded that although there are issues with external validity, the risk of bias and internal validity is generally acceptable.

A funnel plot was generated to assess reporting bias (Section A.3, Figure A1). There appeared to be funnel plot symmetry for in-hospital mortality, although Sterne’s test was not appropriate to detect funnel plot asymmetry due to the small number of studies included in each meta-analysis.

The overall certainty of evidence and the reasons for lowering the ratings are summarised in Table 4. The certainty of evidence was generally low or very low due to the limited availability of RCTs.

## 4. Discussion

The current systematic review revealed a lack of evidence on anaerobic coverage for aspiration pneumonia; only one randomised trial and two observational studies were found eligible for the review. Although very limited in number, these publications did not show a clear benefit of anaerobic coverage in the treatment of patients diagnosed with aspiration pneumonia. No included studies reported benefit of anaerobic coverage in improving mortality.

When considering the need for anaerobic coverage, we must first understand the role of anaerobes in the development of aspiration pneumonia. Two factors, the overestimation and underestimation of their virulence, should be considered.

Previously, the high presence of anaerobes in lower respiratory tract specimens from patients with aspiration pneumonia led to the practice of covering anaerobes for their treatment [4,5,6]. The reported rate of identification of anaerobes in respiratory specimens from patients with aspiration pneumonia was as high as 73.9–100%. Treatment with agents covering anaerobes was often recommended [25,26].

However, since the 1990s, there has been a sharp downward trend in the detection of anaerobes in patients with aspiration pneumonia. The cause of this shift is suspected to be partly due to earlier sampling and intervention. Data reported in the 1970s showing a high prevalence of anaerobic organisms were often derived from samples taken late in the course of the disease [4,5,6,27]. Studies of more acute phase pneumonia have shown less impact of anaerobes [8,28,29,30]. Another consideration is the change in oral hygiene levels over the years. Oral health status is thought to have improved in recent decades, due in part to the advocacy of routine oral care [31]. There have been reports of improvements over the years in general oral status [32], number of missing teeth [33], and toothbrushing frequency [34]. The improvement in oral health is thought to have affected the oral microbiota and the causative organisms of aspiration pneumonia. Other suspected causes include changes in the demographic characteristics of patients [9], as study populations have shifted from relatively young patients with alcoholism or general anaesthesia to older patients. These changes may have contributed to the decrease in anaerobes being identified as pathogens.

Alternatively, despite advances in microbial testing methods, not all pathogens are identified. Anaerobes are known to be difficult to obtain and culture. Therefore, the fact that anaerobes are not identified does not rule out their possibility of being the causative organism. This risk of underestimating the involvement of anaerobes may lead to undertreatment, putting the patient at risk of prolonged illness, treatment failure, and death. Of the three studies included in this review, two used blood/sputum cultures and urine antigens to investigate the causative organism [20,23]. One study did not report any bacteriological analyses [21]. None of the studies reported a method to isolate anaerobes and the results do not mention the identification of anaerobes. Therefore, the risk of underestimating anaerobic involvement must be considered.

Furthermore, the identification of an organism from the respiratory tract does not automatically define it as the cause of active infection. The virulence of anaerobes is not always high [3,4]. Overestimation of microbiological results leads to unnecessary antimicrobial coverage, with the risk of adverse events and complications such as C. difficile infection, and a burden on healthcare costs. In addition, not all anaerobes require additional empirical anaerobic coverage with beta-lactams or clindamycin [30]. The shift in oral anaerobes also suggests that common anaerobes causing aspiration pneumonia may be susceptible to routine CAP treatment [19], although the clinical scenario must also be taken into account [15]. Therefore, even in cases where anaerobes are thought to be the cause of aspiration pneumonia, this does not automatically justify the use of specific antibiotics to cover them.

The definition of aspiration pneumonia is not well established. Although there is a common understanding that aspiration pneumonia is a pneumonia in people with risk factors or signs of aspiration [35], there are no robust criteria. The reported ratio of aspiration pneumonia in community-acquired pneumonia ranges from 5.6% to over 90% [1,36,37] and is highly variable depending on the setting, population, and local understanding of the disease.

Among the studies included in this review, Oi et al. included patients who were diagnosed with CAP/NHCAP who were at risk of aspiration [23], Hasegawa et al. included pneumonia with an aspiration-related risk factor [21], and Marumo et al. included aspiration pneumonia in the NHCAP group B (no risk of MDR pathogen) [20]. The diagnosis of aspiration pneumonia in the presence of one risk factor (such as a history of stroke) is one of the broader definitions [35], compared with others that assess more factors such as swallowing function or pneumonia distribution. If patients with CAP are overly being diagnosed as aspiration pneumonia, this may lead to an underestimation of the role of anaerobes and necessity of anaerobic coverage. While a broad diagnosis of aspiration pneumonia may be meaningful in clinical practice (in order to prevent overlooking the possibility of an aspiration and to assess risk factors and swallowing function carefully), the risk of overdiagnosis cannot be denied in the current context of research. For research purposes, it is necessary to have a common definition of aspiration pneumonia.

Not all risk factors for aspiration are uniformly associated with aspiration pneumonia; rather, the degree to which they cause the disease is thought to vary. For example, in a recent study, of the common risk factors for aspiration pneumonia, impaired consciousness was the most closely associated with chest images suggestive of aspiration pneumonia [38]. Labelling patients with CAP/NHCAP with any risk factor of aspiration as ‘aspiration pneumonia’ may result in the concept of the disease being too broad.

The current suggested approach is to consider aspiration pneumonia not as a distinct entity, but as a continuum of community or hospital acquired pneumonia [9,39,40]. As the associated risk factors and degree of aspiration vary within the spectrum of aspiration pneumonia, the need for anaerobic coverage is also expected to vary.

Current treatment recommendations by various guidelines are highly dependent on observational studies, and varies between regions [17]. The ATS/IDSA guidelines recommend that anaerobic coverage should not be routinely added for suspected aspiration pneumonia unless lung abscess or empyema is suspected [14]. This is mainly based on observational studies reporting a decrease in the detection of anaerobes as causative organisms [28,29,30]. Our systematic review and meta-analysis are in line with these publications, and add interventional evidence to this view.

Our review shows that, according to the current literature, anaerobic coverage may not always be beneficial in the treatment of aspiration pneumonia. Anaerobic coverage may be unnecessary for initial empiric treatment in the absence of abscess formation or empyema and with good oral hygiene. Further management should be based not merely on the diagnostic labelling, but through consideration of patient history, comorbidities, level of consciousness, oral health, previous and current microbiology results, local antibiogram data, previous treatment and nursing/medical care, severity, chest imaging results, and response to treatment [41,42].

There are some limitations in this study that should be mentioned. This review focused on aspiration pneumonia. As the definition varies between settings [35], studies that did not mention the term ‘aspiration’ may not have been identified in the search process. Therefore, we performed manual searches of guidelines and references of relevant papers to identify related articles that may have been missed in the database searches, and added ‘necrotising pneumonia’ and ‘lung abscess’ to our search terms. Despite adding these terms to the search strategy, none of the included studies mentioned whether their participants had necrotising pneumonia or lung abscess. Nevertheless, caution should be taken in interpreting the results, as aspiration pneumonia is still a variable term. Additionally, the included studies all originated from Japan. However, as there was no restriction on the year or country of publication, this is a reflection of the characteristics of the current literature. It is possible that there are differences compared with other countries, although local data do not support this [39]. In addition, the antimicrobials selected in the studies were not uniform. Therefore, no general recommendation can be made from this result. As this is an issue of high clinical importance, further research is needed on the optimal antibiotic treatment of aspiration pneumonia and how to select those who may benefit from anaerobic coverage.

## 5. Conclusions

In the current review, no clear evidence was found to recommend routine anaerobic coverage for the antibiotic treatment of aspiration pneumonia. There are insufficient data to assess the necessity of anaerobic coverage. Further studies are needed to determine which cases of aspiration pneumonia require anaerobic coverage, if any.

## Figures and Tables

**Figure 1 jcm-12-01992-f001:**
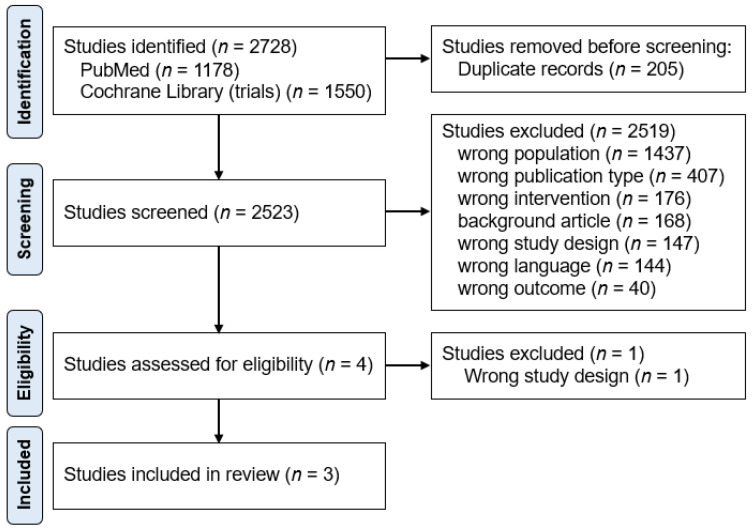
Flow chart of the study process. Through searching databases, 2728 reports were found. After removing duplicates, 2523 reports were screened, of which 2519 were excluded. A total of four studies underwent full-text review, and three studies were included in the review.

**Figure 2 jcm-12-01992-f002:**
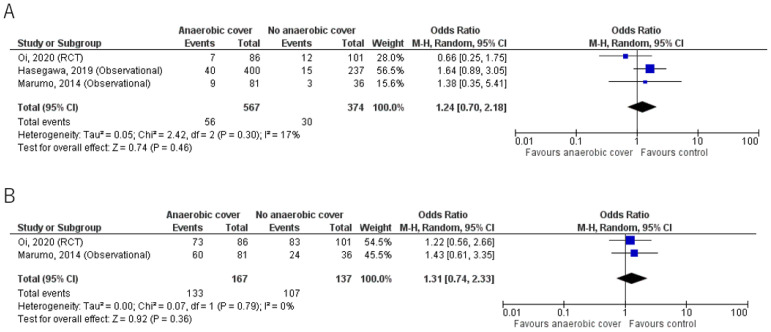
(**A**) Mortality; (**B**) Clinical cure rate. Forest plots comparing outcomes in groups given antibiotic treatment with or without anaerobic coverage. The blue square represents the odds ratios of individual studies. The black diamond represents the pooled result [20,21,23].

**Table 1 jcm-12-01992-t001:** Study Characteristics.

Author, Year	Country	Design	Setting	Subjects	Age (Years)	Antibiotics (Number of Subjects)	
						Anaerobic Coverage Group	Control Group
Oi, 2022 [23]	Japan	Open-labeled Randomized comparative trial	Single centre, inpatient	Moderate to severe CAP/NHCAP patients at risk of aspiration	mean 85	MEPM (86)	CFPM (101)
Hasegawa, 2019 [21]	Japan	Prospective observational	Multicentre, inpatient/outpatient	Pneumonia patients with aspiration-related risk factor	median 77	SBT/ABPC (400)	CTRX (237)
Marumo, 2014 [20]	Japan	Prospective observational	Single centre, inpatient/outpatient	Aspiration pneumonia within the NHCAP group B (no risk of MDR pathogen)	mean 78	SBT/ABPC (81)	AZM (36)

CAP: community acquired pneumonia; NHCAP: nursing-and healthcare-associated pneumonia; MEPM: meropenem; CFPM: cefepime; SBT/ABPC: sulbactam/ampicillin; AZM: azithromycin.

**Table 2 jcm-12-01992-t002:** Mortality and clinical cure rate.

Author, Year	Mortality (30 Day * or in Hospital)			Clinical Cure Rate (*n*, %)		
	Anaerobic Coverage	Control	OR, 95%CI	Anaerobic Coverage	Control	OR, 95%CI
Oi, 2020 [23]	7/86 * (8.1%)	12/101 * (11.9%)	0.66 [0.25, 1.75]	73/86 (84.9%)	83/101 (82.2%)	1.22 [0.56, 2.66]
Hasegawa, 2019 [21]	40/400 (10.0%)	15/237 (6.3%)	1.64 [0.89, 3.05]	NR	NR	NR
Marumo, 2014 [20]	9/81 (11.1%)	3/36 (8.3%)	1.38 [0.35, 5.41]	60/81 (74.1%)	24/36 (66.7%)	1.43 [0.61, 3.35]

OR: odds ratio, CI: confidence interval, NR: not reported.

**Table 3 jcm-12-01992-t003:** Risk of bias assessment, Newcastle–Ottawa Scale.

Author, Year	Selection				Comparability	Outcome			Total Score
	Representativeness of Exposed Cohort	Selection of Controls	Ascertainment of Exposure	Demonstration that Outcome of Interest Was Not Present at Start of Study	Comparability of Cohorts on the Basis of the Design or Analysis	Assessment of Outcome	Adequate Length of Follow-up	Adequacy of Follow up of Cohorts	
Oi, 2020 [23]	b	a	a	a	a	c	a	a	7
Hasegawa, 2019 [21]	b	a	a	a	a	c	a	a	7
Marumo, 2014 [20]	c	a	a	a	a	c	a	a	6

(a, b, and c were allotted according to the criteria as defined by the Newcastle-Ottawa Scale [22]).

**Table 4 jcm-12-01992-t004:** Summary of findings.

Outcomes	No of Participants (Studies)	Odds Ratio [95% CI]	Certainty of Evidence (GRADE)	Reason for GRADing	Comments
Mortality	941 (3)	1.24 [0.70, 2.18]	Very low	Risk of bias, imprecision	There may be little or no difference in the mortality.
Clinical cure rate	304 (2)	1.31 [0.74, 2.33]	Very low	Risk of bias, imprecision	There may be little or no difference in the clinical cure rate.
Development of resistant bacteria	0 (0)	-	-	-	No data available
Length of hospital stay	117 (1)	-	Very low	Risk of bias, imprecision, indirectness	There may be little or no difference in the length of stay.
Recurrence rate	187 (1)	-	Low	Imprecision, indirectness	There may be little or no difference in the rate of recurrence.
Adverse effect rate	176 (1)	-	Low	Imprecision, indirectness	There may be little or no difference in the rate of adverse effects.

CI: confidence interval, GRADE: GRADE Working Group grades of evidence.

## Data Availability

All relevant data are within the manuscript.

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
