# Peer review of "The Clinical Significance of Anaerobic Coverage in the Antibiotic Treatment of Aspiration Pneumonia: A Systematic Review and Meta-Analysis"

_jcm, 2023, doi:10.3390/jcm12051992_

Round 1

Reviewer 1 Report

Yoshimatsu et al. conducted a study titled “The clinical significance of anaerobic coverage in the antibiotic treatment of aspiration pneumonia: a systematic review and meta-analysis”. The authors present an interesting manuscript about an important clinical question. The methods appear appropriate. However, the literature is very limited. I think this is already the primary finding of this study: There is insufficient evidence.

Here are some suggestions

1.     Abstract: Please provide some numbers (even if insignificant) in the abstract. 

2.     Line: Is there are reason why all of these 3 studies were performed in Japan or is this coincidence?

3.     Please perform a sensitivity analysis in which you use the propensity matched data and not the raw data Hasegawa study.

4.     I would slightly change the main conclusion on the discussion and the abstract from “there is no evidence for the benefit of anaerobic coverage” to “there is insufficient data to assess the necessity of anaerobic coverage. 

Reviewer 2 Report

The authors present a systematic review and meta-analysis of studies comparing antibiotics with and without anaerobic coverage in the treatment of aspiration pneumonia. They find that addition of anti-anaerobic antibiotic coverage did not reduce mortality or improve clinical cure rate. The results are in line with current guidelines to avoid anti-anaerobic coverage for uncomplicated aspiration pneumonia. The manuscript is well written and well presented. A number of areas were identified below that could further strengthen the manuscript.

Major Comments:

1.       The authors correctly point out that recent guidelines for treating aspiration pneumonia do not recommend using anti-anaerobic antibiotics and correctly mention that this does not (or should not) apply to patients with empyema or lung abscess, which are commonly caused by anaerobic or mixed aerobic/anaerobic organisms. However, it appears their inclusion criteria for the study included papers with subjects with lung abscess. Can the authors stratify the results by aspiration without lung abscess, and with lung abscess? This should be addressed in the paper.

2.       There is increasing literature suggesting that anaerobic coverage may not only be non-beneficial, but may actually be harmful, for instance, PMID: 36229047 and PMID: 36694848. The authors should mention this literature in the introduction as they frame the question posed by this manuscript.

3.       The authors state in the Introduction that there has never been a review on this topic, however, I do not think this is correct. For instance, see PMIDs: 11228282, 31516051, 25129577. They even cite reviews by Bartlett.

4.       Would the manuscript by Tokuyasu (PMID: 19182422) have met inclusion criteria?

5.       Because only 3 manuscripts were included out of 2728, the authors should list in figure 1 the reasons for exclusion with the N for each reason to clarify why so many were excluded.

6.       The conclusions stated by the authors throughout the manuscript (abstract, discussion, conclusion) should be toned down in light of the fact that the certainty of evidence was low or very low and only 3 studies were able to be included in the analysis.

Minor Comments:

1.       2nd line in abstract – change “is’ to following: “While antibiotics covering anaerobes ARE…”.

2.       The authors repeat that the search was English language only twice in the Methods: Lines 82 and 92.

Round 2

Reviewer 2 Report

The authors have addressed my previous concerns. Two issues identified in the revision:

Necrotising is misspelled line 89

Line 316-318 - I don't understand what this means. Do you mean overestimated role of anaerobes?

Author Response

Dear Reviewer 2,

We appreciate your important input. We have edited the paper accordingly. 

1. Necrotising is misspelled line 89

We have edited the spelling.

2. Line 316-318 - I don't understand what this means. Do you mean overestimated role of anaerobes?

We apologise for the unclear sentence. We meant that overly diagnosing pneumonia as 'aspiration pneumonia' can be a problem in research. This is because if the study participants are too heterogenous and include not only definite aspiration pneumonia but also suspected aspiration pneumonia or pneumonia with minor aspiration factors, it is difficult to assess the necessity of anaerobic coverage etc. For research purposes, it is necessary to have a common definition aspiration pneumonia. We have rephrased this to 'overdiagnosis of aspiration pneumonia', and added the sentence 'For research purposes, it is necessary to have a common definition aspiration pneumonia.' for clarification.

We hope this is a more acceptable form now.

Kind regards,

Kosaku Komiya